# Characteristics Analysis of an Electromagnetic Actuator for Magnetic Levitation Transportation

**Junjie Jin, Xin Wang, Chuan Zhao** **, Fangchao Xu, Wenzhe Pei** ⬛**, Yuhang Liu and Feng Sun \***

School of Mechanical Engineering, Shenyang University of Technology, Shenyang 110870, China
\* Correspondence: sunfeng@sut.edu.cn; Tel.: +86-189-0404-0100

**Abstract:** In this article, an electromagnetic actuator is proposed to improve the driving performance of magnetic levitation transportation applied to ultra-clean manufacturing. The electromagnetic actuator mainly includes the stator with the Halbach array and the mover with a symmetrical structure. First, the actuator principle and structure are illustrated. Afterward, in order to select a suitable secondary structure and analyze the characteristics of the actuator, the electromagnetic characteristics of actuators with different secondary structures are analyzed by the finite element method (FEM). Analysis results show that the actuator adopting the secondary structure with a Halbach array will increase the electromagnetic force and working stability. The secondary with the three-section Halbach array is selected for the electromagnetic actuator. Then, the influence of secondary permanent magnet (PM) thickness on the electromagnetic force is analyzed by FEM. The results indicate that the increase in PM thickness will lead to increased electromagnetic force and decreased utilization ratio of PM. Finally, a prototype of an electromagnetic actuator is built, and experiments are implemented. The correctness of the theoretical analysis and the effectiveness of the electromagnetic actuator are verified by experimental results.

**Keywords:** electromagnetic actuator; Halbach array; characteristic analysis; finite element method



## 1. Introduction

Magnetic levitation is a support technology with no contact and no lubrication. It has been widely applied in the industrial fields, such as maglev trains, precision positioning platforms, and magnetic bearings [1–3]. The development of the semiconductor industry needs more stringent requirements in the manufacturing environment, so magnetic levitation technology has broad application prospects in ultra-clean transportation. In 1998, K.H. Park et al. [4] proposed the maglev conveyor system for ultra-clean manufacturing, which combined AGV (Automated Guided Vehicle) and maglev technology.. Without mechanical contact, the conveyor can effectively improve the air cleanliness in cleanrooms. Moreover, the electromagnetic levitation combination of a planar motor supports rapid responses and high-speed motion [5–8]. They are mainly divided into the moving magnet type and the moving coil type. The moving magnet type has a simple mover structure but requires multiple controllers to achieve precise motion. Conversely, the moving coil type does not require a complex controller. However, serious copper losses cause high power consumption and significant heat emission. Therefore, the cooling mechanism is necessary, which leads to the complex structure and large mass of the magnetic levitation platform. Kim et al. designed a hybrid electromagnetic-permanent magnetic levitation transport system proposed in the reference [9,10]. The magnetic levitation system can realize suspension work with low energy consumption. Furthermore, linear motors were used to drive the magnetic levitation platform. It is impossible to change the levitation gap of the magnetic levitation platform. Moreover, it is harmful to ultra-clean transportation due to dependence on a mechanical guide rail.The permanent magnetic levitation transportation system of variable flux path has low steady-state energy consumption, high controlling

stiffness, and anti-eccentric load characteristics [11,12]. However, a permanent magnetic levitation transport system is sensitive to external disturbance and has high requirements for the stability of the drive system. Magnetic levitation transportation requires a suitable drive system. This drive system has the advantages of lower disturbance, no contact, high precision, and low mass.

The contact drive device is not suitable for magnetic levitation transportation. Electromagnetic drive technology converts electrical energy into mechanical energy, which is the operation of electromagnetic force. It is a contactless drive technology [13,14]. Electromagnetic drive technology has the advantages of fast response, high controllability, and high precision [15–17]. With the development of rare-earth permanent magnetic materials, the electromagnetic drive technology with PM has the advantages of simple structure and high efficiency. Therefore, it is widely researched and applied, such as traffic, delivery platforms, and machine tools [18–20]. Electromagnetic drive technology with PM is categorized into two broad groups: the core type and the coreless type. The iron core type has greater electromagnetic force, but the cogging effect will produce significant disturbances. In addition, the mover and stator have enormous suction, which is detrimental to magnetic levitation transmission [21]. In contrast, the coreless type has less disturbance but less force [22]. Jansen et al. [23] proposed an electromagnetic actuator: a U-shaped stator structure is adopted, and the mover coil is located in the center of the stator. The structure can increase the thrust of the coreless electromagnetic actuator. Furthermore, much research used Halbach arrays for electromagnetic actuators to improve thrust [24–26]. The above research will increase the thrust of the coreless electromagnetic actuator. Moreover, the normal force will increase, and this will increase disturbance. In addition, the normal force was controlled by a highly complex scheme, which caused detrimental effects on the precision. Many accurate control models have been studied to reduce this damage [27–29]. Generally, the above electromagnetic actuator has a minimal air gap (0.3 mm~1 mm). However, the minimal air gap will cause defective effect to the floating of the magnetic levitation platform. In addition, they rely on the guide rail, which limits the application of magnetic levitation transmission.

To make magnetic levitation transportation applicable to the ultra-clean manufacturing environment, an electromagnetic actuator is proposed. The electromagnetic actuator has the advantages of small mass, big electromagnetic thrust, and low disturbance, and can realize automatic guiding used for magnetic levitation transportation. Therefore, it can be combined with a magnetic levitation platform to allow magnetic levitation transport to be used in an ultra-clean manufacturing environment. This paper is organized as follows. Firstly, the electromagnetic actuator principle and structure are illustrated. The objective is to select the appropriate secondary structure and analyze the characteristics of the actuator, therefore, the actuator is presented with three secondary structures of ordinary radial magnetization, a Halbach three-section array, and a five-section array. The resulting models are analyzed by the FEM. Afterward, considering the actuator's electromagnetic characteristics and cost, it determined the three-section Halbach magnet array is an appropriate secondary structure to the electromagnetic actuator. Subsequently, the influence of magnetic thickness on the electromagnetic force of the electromagnetic actuator is analyzed. Finally, a prototype of the electromagnetic actuator is built, and experiments are implemented. The experiment results show the prototype's effectiveness and the analysis's correctness.

## 2. Structure and Principle of Electromagnetic Actuator

### 2.1. Structure and Principle of Halbach Magnet Array

In the Halbach magnet array, the radially magnetized PM plays a dominant role in the magnetic circuit, and the tangentially magnetized PM compensates for the magnetic circuit. The air-gap magnetic field of this PM array tends to sinusoidal distribution, and the harmonic content is less, especially in the continuous magnetization mode, which can produce ideal magnetic field characteristics of the sinusoidal magnetic field waveform.

The Halbach magnet array can enhance the unilateral magnetic field. Figure 1 is the magnetization diagram of the Halbach magnet array, where (a) is a three-section Halbach magnet array, and (b) is a five-section Halbach magnet array. The magnetization angle of two adjacent PMs is $\theta$, and $k$ is the number of PMs contained in a magnetization cycle.

$$\theta = \frac{2\pi}{k} \tag{1}$$

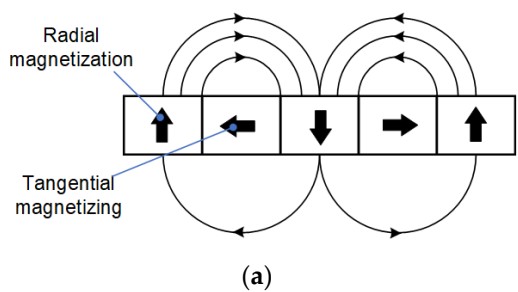
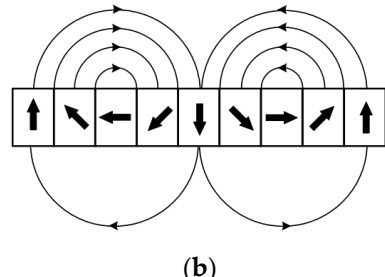

Radial magnetization

Tangential magnetizing

(**a**)　　　　　(**b**)

**Figure 1.** Halbach magnetization diagram: (**a**) Three-section Halbach magnet array; (**b**) Five-section Halbach magnet array.

The Halbach magnet array structure is calculated using the equivalent surface current method based on Ampere's molecular current hypothesis. According to the research results of [30], the magnetic field generated by the PM monomer at any point $p(x, y)$ outside the magnet was expressed as:

$$\begin{cases} B_{1x}(x, y, J) = \frac{\mu_0 J}{4\pi} \ln \frac{(y-h/2)^2 + (x-l/2)^2}{(y+h/2)^2 + (x-l/2)^2} \\ B_{2x}(x, y, J) = -\frac{\mu_0 J}{4\pi} \ln \frac{(y-h/2)^2 + (x+l/2)^2}{(y+h/2)^2 + (x+l/2)^2} \\ B_{1y}(x, y, J) = \frac{\mu_0 J}{2\pi} \left[ \arctan \frac{(y-h/2)}{(x-l/2)} - \arctan \frac{(y+h/2)}{(x-l/2)} \right] \\ B_{2y}(x, y, J) = -\frac{\mu_0 J}{2\pi} \left[ \arctan \frac{(y-h/2)}{(x+l/2)} - \arctan \frac{(y+h/2)}{(x+l/2)} \right] \end{cases} \tag{2}$$

$$\begin{cases} B_x = B_{1x}(x, y, J) + B_{2x}(x, y, J) \\ B_y = B_{1y}(x, y, J) + B_{2y}(x, y, J) \end{cases} \tag{3}$$

where $J$ is the surface current density; $\mu_0$ is the vacuum permeability; $h$ is the PM height; $l$ is the PM width; $B_x$ is the tangential flux density; and $B_y$ is the radial flux density.

According to the coordinate rotation theory, when the PM is tangentially magnetized along the origin, it can be equivalent to the counterclockwise rotation of the coordinate system along the origin by 90°. When the PM is magnetized at any angle $\theta$, the magnetization direction is decomposed into radial and tangential magnetization directions for calculation (where the angle between the magnetization directions and the horizontal direction is $\theta$).

$$\begin{cases} J_{x\theta} = J_\theta \times \cos\theta \\ J_{y\theta} = J_\theta \times \sin\theta \end{cases} , \tag{4}$$

For any group of Halbach magnet arrays, any point $p(x, y)$ magnetic induction intensity distribution can be expressed as

$$\begin{cases} B_x(x, y, J_{x\theta/y\theta}) = \sum_{n=1}^{N} \left[ x - (n - 1/2)l, y - h/2, J_{x\theta/y\theta} \right] \\ B_y(x, y, J_{x\theta/y\theta}) = \sum_{n=1}^{N} \left[ x - (n - 1/2)l, y - h/2, J_{x\theta/y\theta} \right] \end{cases} , \tag{5}$$

### 2.2. Electromagnetic Actuator Structure

Figure 2a is the structure of the electromagnetic actuator. It consists of secondary PM and mover primary winding, in which the actuator bilateral primary symmetrical installation adopts a conjugate structure and the secondary adopts the Halbach magnet array. The primary yoke of the mover adopts non-metallic materials to reduce the heat generated by the actuator due to the eddy current effect during operation. This structure is shown in Figure 2b. In order to compensate for the low density of the electromagnetic thrust generated by the coreless structure, yoke iron is added to the back of the winding. The primary employs a fractional-slot concentrated winding distribution and a 12-slot 10-pole structure, and the winding on both sides are in reverse series. The schematic diagram of its electromagnetic structure is shown in Figure 3. The primary stage adopts coreless armature structure, which eliminates the cogging effect. The bilateral mover can stabilize the bilateral air gap by the normal force of the same size and the opposite direction. Therefore, the structure is applied to the magnetic levitation transport system and can work without relying on the guide rail.

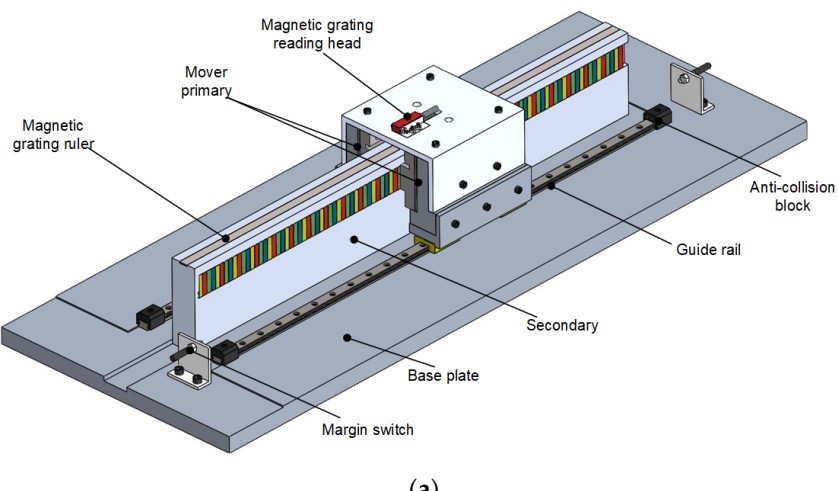

(**a**)

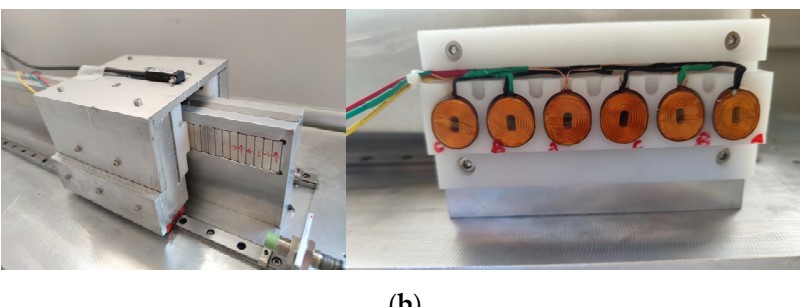

(**b**)

**Figure 2.** Structure diagram of electromagnetic actuator: (**a**) Mechanical structure; (**b**) Mover primary structural.

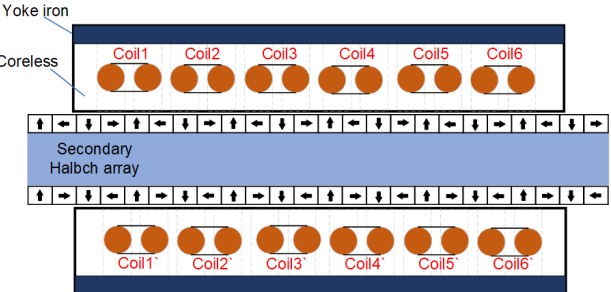

**Figure 3.** Electromagnetic structure of the electromagnetic actuator.

### 2.3. Electromagnetic Actuator Heterogeneous Secondary Structure

Figure 4 shows the cross-section diagram of (a) radial magnetization, (b) a Halbach three-section magnet array magnetization, and (c) a Halbach five-section magnet array magnetization as secondary for the electromagnetic actuator. They have the same slot to pole ratio. The main parameters of the actuator are shown in Table 1.

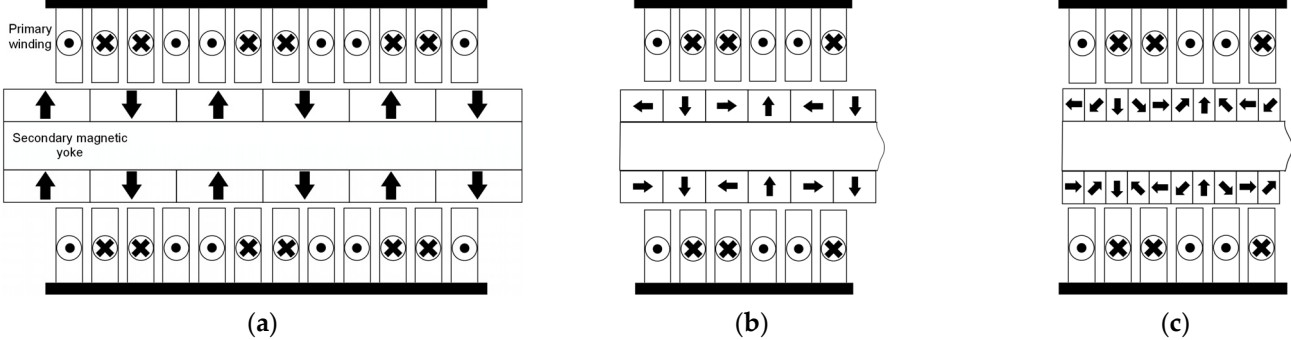

(**a**)      (**b**)      (**c**)

**Figure 4.** Cross-section diagram of three secondary structures: (**a**) Structure of electromagnetic actuator with radially magnetized secondary; (**b**) Structure of electromagnetic actuator with Halbach three-section magnetized secondary; (**c**) Structure of electromagnetic actuator with Halbach five-section magnetized secondary.

**Table 1.** The parameters of the electromagnetic actuator.

| Parameters | Radial Magnetization | Halbach Three-Section | Halbach Five-Section |
|---|---|---|---|
| Pole pitch (mm) | 12 | 12 | 12 |
| Magnet height (mm) | 5 | 5 | 5 |
| Magnet width (mm) | 12 | 6 | 3 |
| Primary winding height (mm) | 19 | 19 | 19 |
| Primary winding width (mm) | 127 | 127 | 127 |
| Winding turns | 489 | 489 | 489 |
| Current (A) | 5 | 5 | 5 |
| gap length (mm) | 2 | 2 | 2 |
| PM | NdFb54 | NdFb54 | NdFb54 |

## 3. Characteristic Analysis of Electromagnetic Actuator

This paper compares the electromagnetic characteristics of the electromagnetic actuator with the same pole to slot ratio which has different secondary structures. The electromagnetic actuator mainly includes the bilateral U-type actuator's primary (mover) and secondary (stator), as shown in Figure 1. The secondary stator is located in the middle of the primary bilateral mover, forming the air gap on both sides, and the air gap on both sides is 2 mm. The tree different secondary structures are a radially magnetized magnet array structure, a three-section Halbach magnet array structure, and a five-section Halbach magnet array structure. Electromagnetic actuator electromagnetic characteristics of three secondary structures are analyzed using the two-dimensional time-step finite element method. In practical application, the adopted joint action of multiple primary units and the cooperation of each unit actuator can be considered as the infinite primary length of the actuator. The end effect caused by the start of each actuator unit has little impact on the electromagnetic characteristics of the drive [31]. Therefore, this paper ignores the influence of the actuator end effect on the actuator.

FEM model sizes for three electromagnetic actuators are shown in Table 1. Silicon steel is used as yoke material with a thickness of 3 mm. The winding material property is copper, and aluminum is used by secondary permanent magnet yoke. In the PM and winding area, the mesh is divided into 0.1 mm. The primary yoke and permanent magnet yoke area

divided the mesh into 0.3 mm, and another area mesh is 0.5 mm. Firstly, the electromagnetic actuator finite element model is divided into two regions to solve, respectively. Then, the solution results of the two regions are synthesized and compared with the complete solution results. The calculation is based on the infinite boundary condition.

### 3.1. FEM of Magnetic Field Analysis

The no-load magnetic field distributions of three different secondary structures of electromagnetic actuators are shown in Figure 5. The (a) shows that the magnetic field distribution generated by the PM is the same on the working and yoke magnet sides. Furthermore, there is more significant magnetic field intensity between adjacent PM s. Therefore, the electromagnetic actuator of the radially magnetized secondary generates significant flux leakage on the contact part of two adjacent PMs and magnet yoke. Conversely, the electromagnetic actuator of the Halbach magnetized secondary has significant magnetic field intensity on the working sides. Because the Halbach magnet can enhance the magnetic field intensity of the working side, the magnetic flux leakage is less. Compared with the magnetic field distributions of two Halbach magnetized secondaries, the magnetic field distribution is more uniform for the five-section Halbach magnet array, and the magnetic flux leakage is the least.

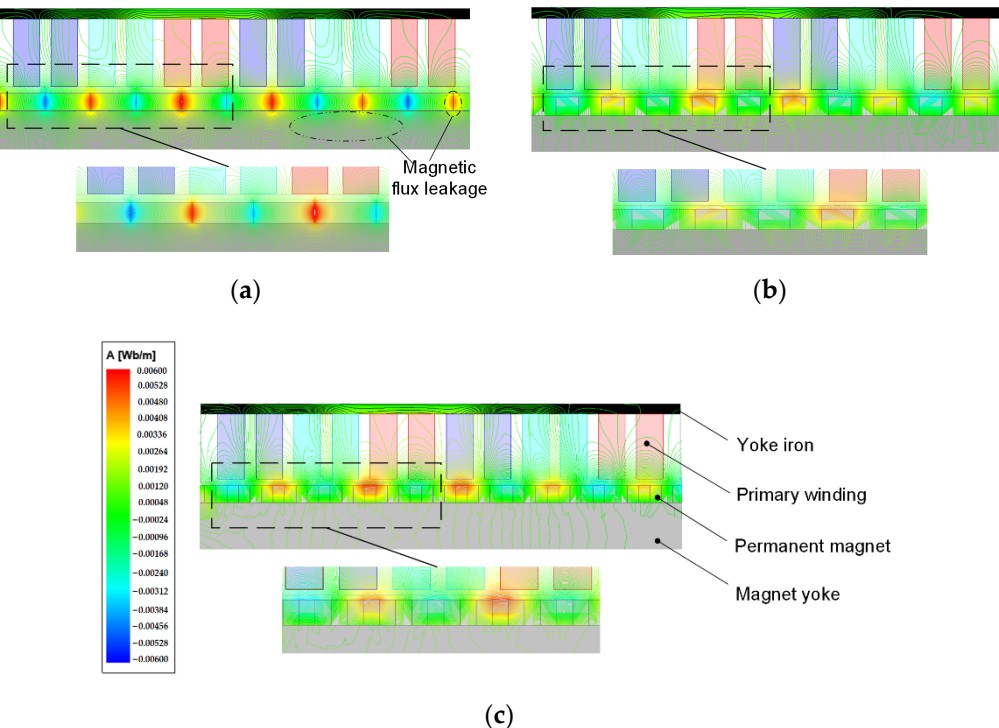

**Figure 5.** Magnetic field distribution: (**a**) Electromagnetic actuator with radially magnetized secondary; (**b**) Electromagnetic actuator with Halbach three-section magnetized secondary; (**c**) Electromagnetic actuator with Halbach five-section magnetized secondary.

Figure 6 shows the magnetic flux density distribution of the horizontal midline of the upper air gap. The Halbach magnet array combination can enhance the air gap magnetic field. Additionally, the more magnets at a pole pitch, the better the sinusoidal air gap flux density. According to Formula (5), the flux density peak at the center of the air gap is calculated. The comparison results are shown in Table 2. The simulated calculation does not consider magnetic flux leakage and assumes that the magnetic field is evenly distributed in the air gap. The Electromagnetic actuator of the radially magnetized secondary has severe magnetic flux leakage, so there are significant differences between the simulation and FEM data. Conversely, the electromagnetic actuator of the Halbach-magnetized secondary is similar between simulation and FEM data.

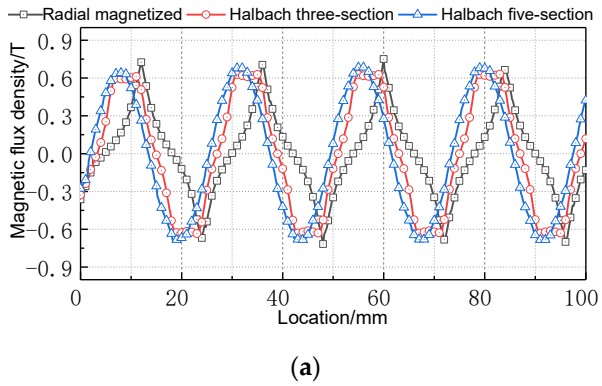
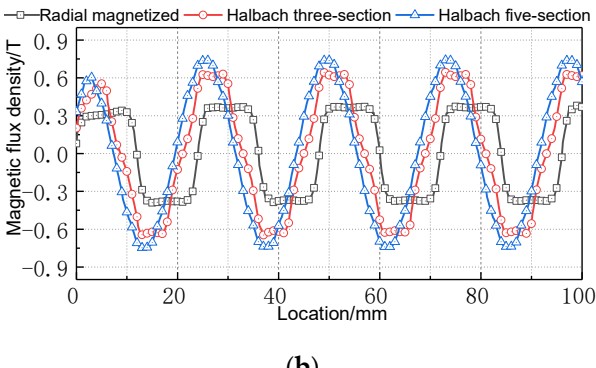

**(a)**                    **(b)**

**Figure 6.** The magnetic flux density of air gap centerline: (**a**) Tangential magnetic flux density; (**b**) Radial magnetic flux density.

**Table 2.** Results of simulation data and FEM data.

| Magnetic Flux Density | | Radial Magnetization | Halbach Three-Section | Halbach Five-Section |
|---|---|---|---|---|
| Tangential magnetic flux density (T) | Simulation | 0.5 | 0.6 | 0.55 |
| | FEM | 0.7 | 0.61 | 0.58 |
| | deviation | 40% | 1.6% | 5.4% |
| Radial magnetic flux density (T) | Simulation | 0.29 | 0.6 | 0.75 |
| | FEM | 0.37 | 0.62 | 0.76 |
| | deviation | 28% | 3.3% | 1.3% |

### 3.2. Electromagnetic Actuator Electromagnetic Thrust Analysis

Electromagnetic thrust is one of the essential parameters of the electromagnetic actuator. This section analyzes the electromagnetic actuator when the primary size is the same different secondary structure. The effects of different secondary structures, namely radial magnetization, the three-section Halbach magnet array, and the five-section Halbach magnet array, on the electromagnetic actuator's electromagnetic thrust performance are compared.

Figure 7 shows the electromagnetic thrust generated by three different secondary structure actuators when the three-phase winding current is 5 A, and the frequency is 10 Hz. When a U-type symmetrical structure is used, the thrust waveform and phase of bilateral primary are the same. Figure 7a is the unilateral electromagnetic thrust and (b) is the regional thrust of (a). Because Figure 7b is a regional thrust enlarged diagram, the thrust fluctuation of the three electromagnetic actuators can be clearly shown. However, the peak and valley values of the overall thrust range cannot be shown in Figure 7b. The calculation formula for the electromagnetic thrust fluctuation is [32]:

$$K_F = \frac{F_{max} - F_{min}}{F_{avg}} \times 100\%, \tag{6}$$

where $F_{max}$ is the peak electromagnetic thrust; $F_{min}$ is the valley value of electromagnetic thrust; $F_{avg}$ is the average electromagnetic thrust.

After analysis and calculation, the same total PM volume, the electromagnetic actuator of radially magnetized secondary electromagnetic thrust peak value is 21.62 N, the valley value is 21.1 N, the average thrust is 21.42 N, and the thrust fluctuation is 2.4%. For the electromagnetic actuator of the Halbach three-section magnetized secondary, the electromagnetic thrust peak value is 34.83 N, the valley value is 34.5 N, the average thrust is 34.76 N, and the thrust fluctuation is 0.84%. For the electromagnetic actuator of the Halbach five-section magnetized secondary, the electromagnetic thrust peak value is 30.84 N, the valley value is 30.71 N, the average thrust is 30.8 N, and the thrust fluctuation is 0.41%. The utilization ratio of PM materials is defined as the ratio of the actuator's average thrust to the

PM's volume. This index measures the utilization of PM materials with high cost. According to this, the PM utilization rate of the radial magnetization secondary is 0.67 N/cm$^3$. The PM utilization ratio of the Halbach three-section magnet array secondary is 1.10 N/cm$^3$. The PM utilization ratio of the Halbach five-section magnet array secondary is 0.978 N/cm$^3$.

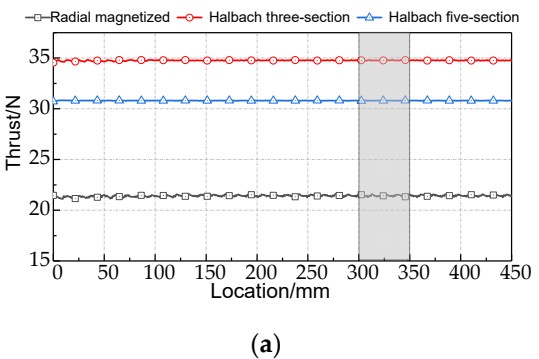

(**a**)

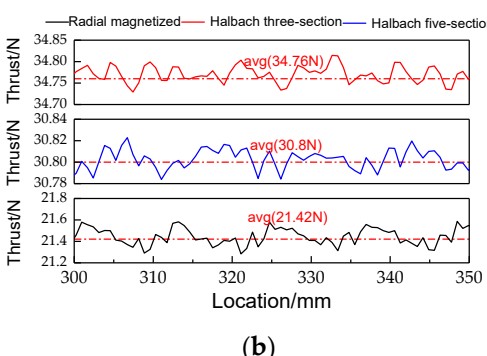

(**b**)

**Figure 7.** Electromagnetic thrust: (**a**) Bilateral combined electromagnetic thrust; (**b**) Regional thrust.

The above analysis shows that the PM utilization rate of the electromagnetic actuator with Halbach three-section magnet array secondary structure is the highest. The more PM blocks at a pole pitch, the smaller the thrust fluctuation and the more stationary the actuator works.

### 3.3. Electromagnetic Actuator Normal Force Analysis

The electromagnetic actuator will generate electromagnetic thrust and normal force. In order to improve the operation accuracy of the actuator, it is necessary to consider the influence of friction disturbance caused by periodic fluctuation of normal force on the horizontal electromagnetic thrust [33].

Figure 8 shows the electromagnetic force results of three different secondary structures for the electromagnetic actuator, when the three-phase winding current is 5 A and the frequency is 10 Hz.

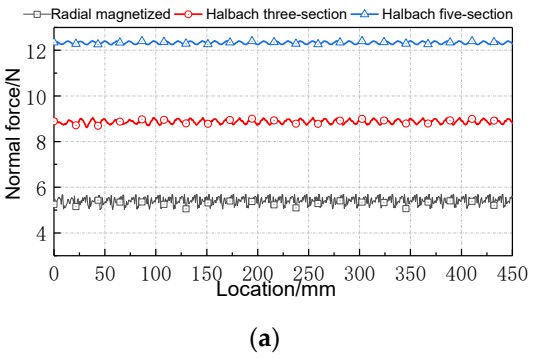

(**a**)

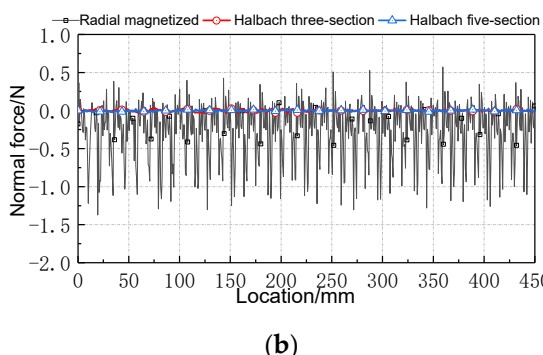

(**b**)

**Figure 8.** Normal electromagnetic force: (**a**) Unilateral normal force; (**b**) Bilateral combined normal force.

After analysis and calculation, the unilateral normal force of the electromagnetic actuator with the same total PM volume is shown in Figure 8a. In the electromagnetic actuator of the radially magnetized secondary, the normal force peak value is 5.69 N, the valley value is 5.02 N, the average normal force is 5.41 N, and the normal force fluctuation is 12%. In the electromagnetic actuator of the Halbach three-section magnetized secondary, the normal force peak value is 9.15 N, the valley value is 8.62 N, the average normal force is 8.88 N, and the normal force fluctuation is 6%. In the electromagnetic actuator of the Halbach five-section magnetized secondary, the normal force peak value is 12.41 N, the

valley value is 12.25 N, the average normal force is 12.32 N, and the normal force fluctuation is 1%.

Electromagnetic actuator adopts symmetrical bilateral primary, so the primary on both sides generates equal normal force with opposite direction. It is specified that the normal force generated on the upper side is negative and on the lower side is positive. The bilateral combined normal force is shown in Figure 8b. In the electromagnetic actuator of the radially magnetized secondary, the normal force peak value is −1.37 N, the valley value is 0.57 N, and the average normal force is −0.4 N. In the electromagnetic actuator of the Halbach three-section magnetized secondary, the normal force peak value is −37 mN, the valley value is 35 mN, and the average normal force is 1 mN. In the electromagnetic actuator of Halbach five-section magnetized secondary, the normal force peak value is 27.6 mN, the valley value is 26 mN, and the average normal force is 0.8 mN.

The above analysis shows that the Halbach magnet array will enhance the radial air gap flux density according to Section 3.1. The normal force increases with the increase of the normal component of air gap flux density. The normal force generated by the bilateral winding of the U-type structure can be offset by each other. The more PM blocks will have minor normal force fluctuation. The smaller the influence of the normal force of bilateral primary synthesis on the thrust fluctuation, the more stable the electromagnetic actuator will be.

### 3.4. Electromagnetic Actuator No-Load Back EMF and Self-Inductance Analysis

The no-load back EMF is one of the parameters that needs to be considered in the design of the electromagnetic actuator, which has an important influence on the static and dynamic performance of the electromagnetic actuator. The three-phase no-load EMF of the electromagnetic actuator with different secondary structures at the synchronous speed of 0.24 m/s is shown in Figure 9. Each phase no-load back EMF is 120° mutual difference. For the electromagnetic actuator with the radially magnetized secondary, the no-load back EMF amplitude is 0.39 V. For the electromagnetic actuator with the Halbach three-section magnetized secondary, the no-load back EMF amplitude is 0.57 V. For the secondary of the electromagnetic actuator with the Halbach five-section magnetized, the no-load back EMF amplitude is 0.61 V. The no-load back EMF sinusoidal waveform quality of the Halbach magnet array secondary structure is higher than that of the actuator with the radial magnetization secondary structure. The no-load back EMF of the actuator with the Halbach five-section magnet array magnetized secondary is the largest, and the sinusoidal waveform is the best.

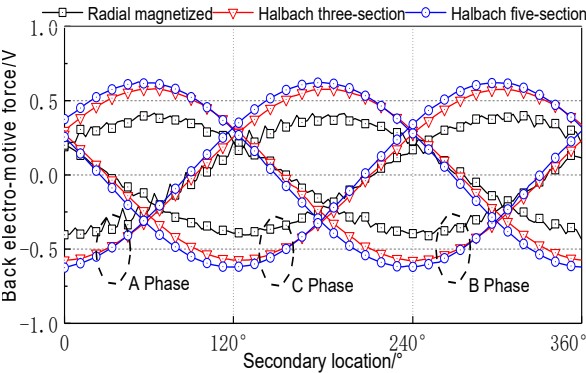

**Figure 9.** Three-phase no-load back EMF.

Electromagnetic actuator self-inductance is one of the critical parameters in actuator design. The self-inductance fluctuation will produce thrust fluctuation, which will harm the thrust fluctuation of the actuator. It has an important effect on the static and dynamic performance of the actuator. Figure 10 shows the actuator's self-inductance curves of A phase primary winding with three different secondary structures. When the primary

winding of the A phase moves to the position with the maximum PM flux interlinkage, the self-inductance is the smallest. When the primary winding of the A phase moves to the minimum position of PM flux interlinkage, the self-inductance is the largest. The self-inductance varies periodically with pole pitch.

Figure 10a is the self-inductance of the electromagnetic actuator with a radially magnetized secondary. The average self-inductance of unilateral A phase winding is 3.663 mH, and the change rate is 0.12%. The Figure 10b is the self-inductance of the electromagnetic actuator with a Halbach three-section magnetized secondary. The average self-inductance of unilateral A phase winding is 3.664 mH, and the change rate is 0.1%. The Figure 10c is the self-inductance of electromagnetic actuator with a Halbach five-section magnetized secondary. The average self-inductance of unilateral A phase winding is 3.663 mH, and the change rate is 0.08%. The self-inductance amplitude and phase of the A phase winding on both sides are the same for the electromagnetic actuator. The self-inductance is superimposed after the series connection. The average self-inductance of the superimposed electromagnetic actuator of the radially magnetized secondary is 7.327 mH, and the change rate is still 0.12% as that of the unilateral. For the electromagnetic actuator of the Halbach three-section and five-section magnetized secondary, the average self-inductance of A phase winding is 7.328 mH and 7.321 mH, respectively, and the change rate is the same as that of a unilateral.

According to the above analysis, the electromagnetic actuator by the Halbach magnet array secondary structure self-inductance waveform is closer to the sinusoidal waveform. The Halbach magnet array secondary structure generation change rate of self-inductance is lower. It is further explained that the Halbach magnet array type secondary can reduce the thrust fluctuation. The pole pitch has more PM blocks, and the actuator thrust fluctuation will be smaller.

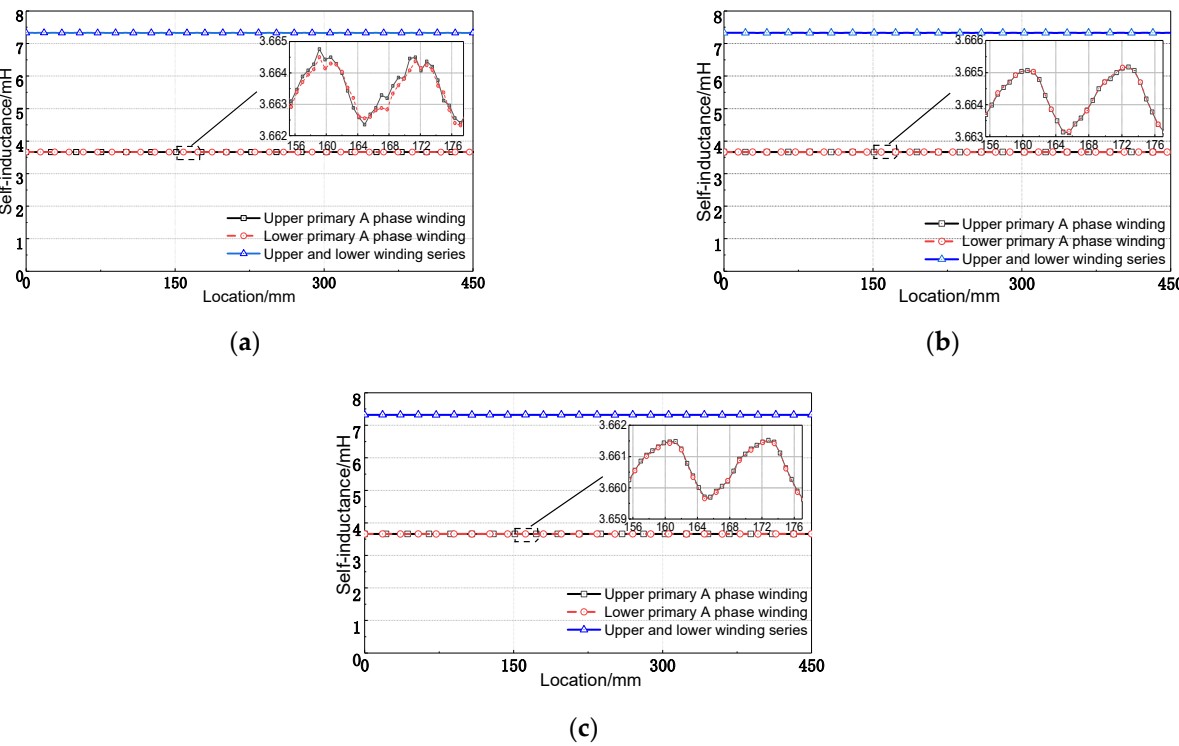

**Figure 10.** Electromagnetic actuator self-inductance: (**a**) Electromagnetic actuator with radially magnetized secondary; (**b**) Electromagnetic actuator with Halbach three-section magnetized secondary; (**c**) Electromagnetic actuator with Halbach five-section magnetized secondary.

Through the analysis of Sections 3.1–3.4, the performance comparison parameters of three different electromagnetic actuators are obtained, as shown in Table 3.

**Table 3.** Comparison of different electromagnetic actuators.

| Parameters | Radial Magnetization | Halbach Three-Section | Halbach Five-Section |
| --- | --- | --- | --- |
| Average thrust (N) | 21.42 | 34.76 | 30.8 |
| Average normal force (N) | −0.4000 | 0.0010 | 0.0008 |
| No-load back EMF (V) | 0.39 | 0.57 | 0.61 |
| Thrust fluctuation | 2.4% | 0.84% | 0.41% |
| PM utilization ratio (N/cm$^3$) | 0.67 | 1.10 | 0.97 |
| Self-inductance fluctuation | 0.12% | 0.1% | 0.08% |

## 4. Analysis of the Influence of PM Thickness on Electromagnetic Force

Compared to electromagnetic actuators with three-section and five-section, both disturbances can be accepted by magnetic levitation transportation. Furthermore, electromagnetic actuator with Halbach three-section magnetized secondary has significant thrust and the PM utilization ratio. Moreover, it also possesses the low cost.The Halbach three-section magnetized secondary is selected for the electromagnetic actuator. The pole size is shown in Figure 11. Pole pitch $\tau$ is defined as the distance between two proximity radially magnetized centers. The $\tau$ = 12 mm. When the single magnet width $w$ = 12 mm, the influence of 3~12 mm thickness $h$ on actuator performance is analyzed.

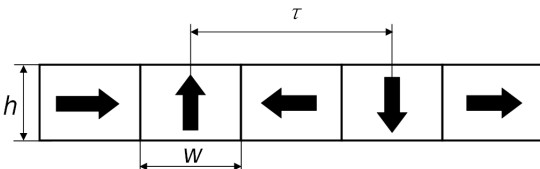

**Figure 11.** Schematic diagram of pole size.

Figure 12 shows the average electromagnetic thrust of the electromagnetic actuator with PM thickness $h$. The figure shows that the average electromagnetic thrust increases with the increase of PM thickness $h$ but the curve slope of the average electromagnetic thrust decreases. After analysis, the electromagnetic actuator tangential air gap flux density increases with the magnet thickness. When the thickness is $h$ > 9 mm, the growth trend of tangential magnetic density tends to be gradual. The utilization rate of the PM is shown in Table 4. When the secondary pole pitch $\tau$ of the Halbach magnet array type is constant, the utilization ratio of the PM decreases with the increase of magnet thickness $h$.

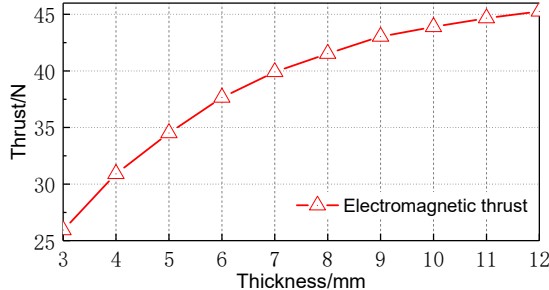

**Figure 12.** The curve of thrust variation with $h$.

Figure 13 shows the variation curves of the unilateral average normal force and the bilateral combined average normal force of the electromagnetic actuator with the magnetic pole thickness $h$. The relationship between radial magnetic density and magnet thickness is the same as tangential magnetic density. So, the average normal force also increases with the magnetic pole thickness $h$. Like the electromagnetic thrust, the average normal force's curve slope becomes smaller. Since the U-shaped symmetrical structure is adopted,

the normal force of the bilateral synthesis tends to zero. Therefore, the normal force is insensitive to the change in magnet thickness.

**Table 4.** PM utilization ratio of different magnet thicknesses.

| Thickness (mm) | PM Utilization Ratio (N/cm$^3$) |
|:---:|:---:|
| 3 | 1.362 |
| 4 | 1.215 |
| 5 | 1.103 |
| 6 | 0.987 |
| 8 | 0.817 |
| 9 | 0.753 |
| 10 | 0.691 |
| 11 | 0.639 |
| 12 | 0.595 |

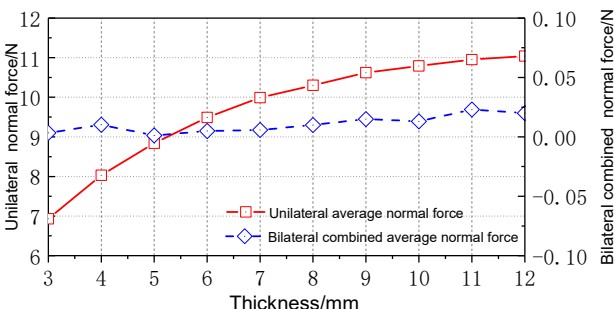

**Figure 13.** The curve of normal force variation with *h*.

## 5. Experimental Validations

To verify the correctness of the above results, design an electromagnetic actuator prototype with a three-section Halbach magnet array secondary structure, as shown in Figure 14. The prototype parameters are shown in Table 5. The prototype's no-load back EMF and the average thrust of different current excitations are tested. The FEM value is consistent with the measured value of the prototype. Still, the FEM value is slightly higher than the measured value because the two-dimensional finite element calculation fails to consider the influence of the transverse end effect and the end magnetic flux leakage.

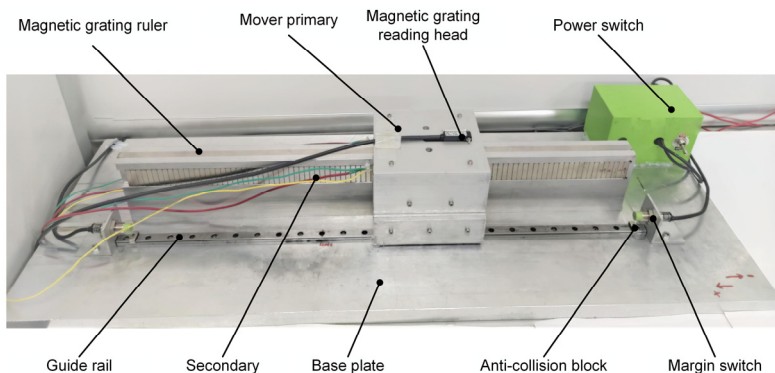

**Figure 14.** Electromagnetic actuator prototype.

Figure 15 is the no-load back EMF test platform. The linear module is controlled by PLC, which drives the prototype to move synchronously at a speed of 0.24 m/s. The no-load back EMF of the prototype at a speed of 0.24 m/s can be observed from the oscilloscope. The no-load back EMF of the electromagnetic actuator prototype at the synchronous speed of 0.24 m/s is shown in Figure 16. Each phase no-load back EMF is 120° mutual difference,

the amplitude is 0.53 V, and they have better sinusoidal waveform. Compared with other windings, the B-phase winding has less amplitude than FEM. This error is due to the large installation gap between the stator and the B-phase winding coil.

**Table 5.** The parameters of the electromagnetic actuator prototype.

| Parameter | Value |
|---|---|
| Mover mass (Kg) | 2.99 |
| Secondary length (mm) | 600 |
| Magnet height (mm) | 5 |
| Pole pitch (mm) | 12 |
| Mover length (mm) | 127 |
| Mover width (mm) | 109 |
| Distance of move (mm) | 460 |
| Resistance ($\Omega$) | 7.1 |
| Gap length (mm) | 2 |
| Pole pitch (mm) | 12 |

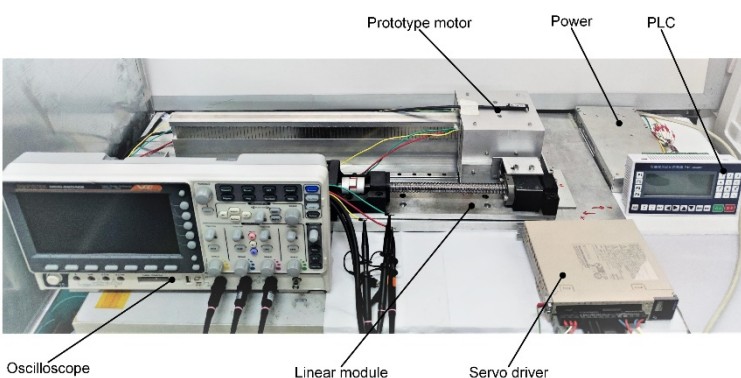

**Figure 15.** The no-load back EMF test platform of the prototype.

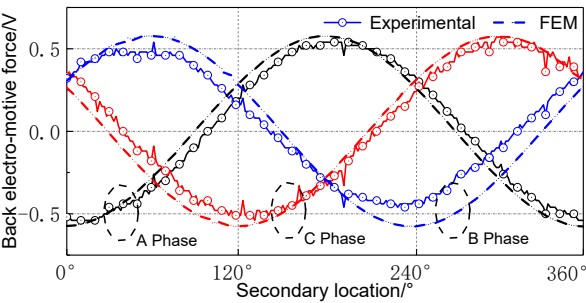

**Figure 16.** No-load back EMF.

Figure 17 is the electromagnetic thrust test platform of the prototype. Thrust measurement tests under different current excitations were performed Firstly, one side of the force sensor is fixed on the stator, and the other is fixed on the mover. Secondly, the prototype is set to the current control mode by the computer (PC). Thirdly, the PC gives the servo drive different current signals, and the servo drive controls the prototype to work with different currents. Finally, the electromagnetic thrust generated by the prototype at different currents is displayed by the force sensor. The unilateral and bilateral electromagnetic thrusts are measured. The measured values of the prototype compared with FEM are shown in Figure 18: (a) unilateral electromagnetic thrust and (b) bilateral electromagnetic thrust. When the current is less than 1.3 A, the electromagnetic force generated by the prototype is less than the starting thrust, and the measured thrust does not match the FEM value. When the current is more than 1.3 A, the measured thrust of the prototype is consistent with the FEM value.

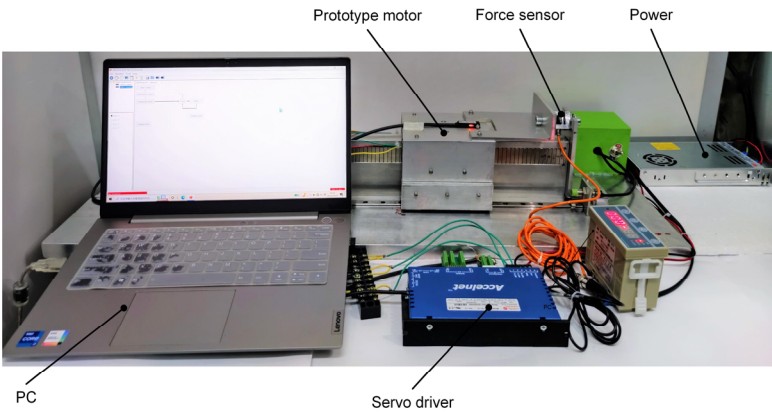

**Figure 17.** The electromagnetic thrust test platform of the prototype.

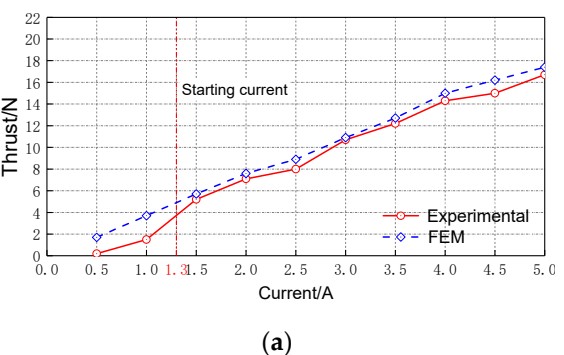

(**a**)

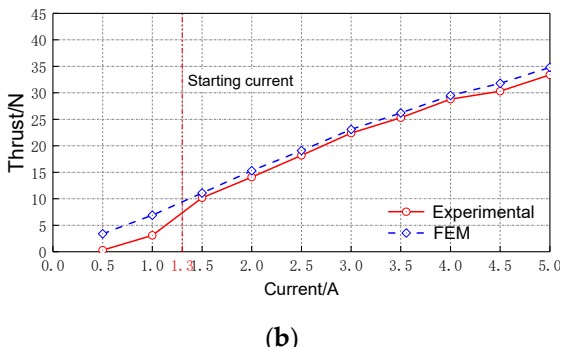

(**b**)

**Figure 18.** Electromagnetic thrust: (**a**) Unilateral electromagnetic thrust; (**b**) Bilateral electromagnetic thrust.

## 6. Conclusions

This paper proposes a self-direction electromagnetic actuator applied in magnetic levitation transportation. Electromagnetic actuator characteristics of different secondary structures are analyzed by the FEM. This paper explained the electromagnetic actuator has a big electromagnetic thrust and low disturbance. Afterward, the Halbach three-section magnet array secondary was selected as the secondary structure of the electromagnetic actuator. This structure's secondary thickness's influence on the electromagnetic force was further analyzed. Finally, the experimental prototype was made to verify the correctness of the analysis and effective application to magnetic levitation transport systems. The conclusions are as follows:

(1) For the electromagnetic actuator with the Halbach three-section magnet array secondary compared with the radial magnetization secondary, the electromagnetic thrust improved by 62.1%, the normal force reduced from 0.4 N to 1 mN, and the thrust fluctuation reduced from 2.4% to 0.84%. Moreover, for the secondary structure of the Halbach magnet arrays, the PM utilization ratio improved by 64.2%. This shows the Halbach magnet array can effectively improve the electromagnetic thrust and reduce the disturbance.

(2) Comparing the electromagnetic actuator of the Halbach three-section magnet array secondary and the Halbach five-section magnet array secondary, the Halbach three-section magnet array secondary electromagnetic thrust improved by 12.9%; the normal force improved from 0.8 mN to 1 mN; the thrust fluctuation improved from 0.41% to 0.84%; and the PM utilization ratio improved by 13.4%. The above description electromagnetic actuator by Halbach three-section magnet array secondary has a larger electromagnetic thrust and a higher PM utilization ratio. However, the stability of the motion will be reduced in comparison to the electromagnetic actuator of the Halbach five-section magnetized secondary. In addition, the Halbach five-section magnetized secondary has a high cost. Thus, the Halbach three-section magnet array is more suitable for practical application.

(3) The air gap flux density increases with the thickness of the PM. Hence, the electromagnetic thrust and normal force increases with the thickness. The PM utilization ratio decreases with increasing thickness.

**Author Contributions:** Conceptualization, J.J., X.W. and C.Z.; methodology, F.S., X.W., C.Z. and J.J.; software, J.J., X.W. and F.X.; validation, F.S., C.Z., W.P., Y.L. and J.J.; formal analysis, F.S., F.X., W.P. and J.J.; investigation, F.S., X.W., F.X. and J.J.; resources, F.S., J.J. and F.X.; data curation, X.W. and W.P.; writing—original draft preparation, J.J., X.W., F.S., Y.L. and C.Z.; writing—review and editing, J.J., X.W., and F.S.; visualization, F.S., X.W., F.X. and J.J.; supervision, F.S. and J.J.; project administration, F.S.; funding acquisition, F.S. All authors have read and agreed to the published version of the manuscript.

**Funding:** This research is supported by National Natural Science Fund of China (No.52005345, No. 52005344), National Key Research and Development Project (No.2020YFC2006701), Scientific Research Fund Project of Liaoning Provincial Department of Education (No. LFGD2020002), Major Project of the Ministry of Science and Technology of Liaoning Province (No.2022JH1/10400027).

**Conflicts of Interest:** The authors declare no conflict of interest.

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
