# Peer review of "Characteristics Analysis of an Electromagnetic Actuator for Magnetic Levitation Transportation"

_actuators, doi:10.3390/act11120377_

Round 1

Reviewer 1 Report

The paper analyzes characteristics of an electromagnetic actuator with linear thrust for magnetic levitation transport. Some experiments are carried out to veirfy the halbach configuration.

1. I can't see the specific research purpose and significance in introduction. Can you highlight the main contributions of the paper?

2. What are the merits of the proposed electromagnetic actuator when it is used in the magnetic levitation system? Its thrust increases but the accuracy decreases.

3. How is the thrust measured in the experiment? Can you introduce the process of thrust measurement.

4. Figs 1,10,11,14,15,17 are very shadowy. You should improve the quality of photos.

Author Response

Dear reviewer:

Best wishes

Reviewer 2 Report

1.           Fig.1(a) and Fig.10 are not clear.

2.           What do the different colors in Fig.5 mean?

3.           It is recommended to provide a table to summarize the comparison results between different secondary structures.

4.       It seems that the comparison of different secondary structures is one of the main contributions of this paper. It is better to add the comparison results to the abstract part.

5.       Similarly, the specific analysis conclusion of the influence of secondary permanent magnet size on the electromagnetic force should also be added to the abstract part.

6.       English expressions should be reviewed and checked carefully.

Author Response

Dear reviewer

Best wishes

Reviewer 3 Report

given comments to Authors

1.      The paper gives a linear electromagnetic actuator structure using a Halbach array. The actuator structure is given at the beginning of the paper, which should be shifted to the next part. The analytical analysis and theoretical background should be given in the first part.

2.      The FFM analysis must be done carefully in a separate section. The magnetic field distribution is presented poorly. There is no legend data and legend with flux density.

3.      Also, is not clear what are the simulation data and FFE model data.

4.      The flux losses analysis using FFM is missed.

5.      The actuator structure and mechanical/electrical data are missing.

6.      Given results are questionable and hard to verify. Also, some Figures are not readable. i.e. Fig. 10.

7.      Moreover, some results need comments, i.e. differences between FEM and tests.

8.      Electromagnetic dynamics is known as a strongly nonlinear system with structured uncertainty. These are not addressed.

9.      The set-ups need more details.

10.    The conclusions should be supported by results.   I invite Authors to respond to the reviewer’s comments and revise their papers.

Author Response

Dear reviewer

Best wishes

Reviewer 4 Report

The subject of the work is current and worthy of interest. Transport on a magnetic cushion is an intensively developed technology. Any research that can bring more knowledge about the processes in the drive itself and around the vehicle is valuable.

The presented research position is entirely original, but its presence in the work is too general. It is necessary to improve the quality of figures: 1, 2, 3, 4, 10, 11, 14, 15, and 17. Their readability is severely limited.

The review of the literature is relatively modest. 23 references, of which only 12 are new (2020-2022). In addition, almost 100% of the literature consists of works from only one geographical area. Scientific research in this area is conducted all over the world. Limiting the authors only to the achievements of one area is inappropriate.

Please prepare an abstract in accordance with the guidelines of the journal. According to the shema applicable in the MDPI publication: background, methods, results, conclusions. Authors have 200 words at their disposal, so make the most of them. 

Unfortunately, the work lacks a discussion of the achievements of other authors, which is unacceptable in scientific works. Please provide a discussion.

In general, the work has great cognitive potential but requires major revision.

Author Response

Dear reviewer

Best wishes

Round 2

Reviewer 1 Report

no comments

Reviewer 3 Report

thank you, I have no more comments

Reviewer 4 Report

The changes introduced in the content of the article allow for a positive assessment of the work. I recommend it for publication.